# Markovian to non-Markovian phase transition in the operator dynamics of a mobile impurity

Dominic Gribben[1], Jamir Marino[1] and Shane P. Kelly[1,2]⋆

**1** Institute for Physics, Johannes Gutenberg University of Mainz, D-55099 Mainz, Germany
**2** Mani L. Bhaumik Institute for Theoretical Physics, Department of Physics and Astronomy, University of California at Los Angeles, Los Angeles, CA 90095, USA

⋆ skelly@physics.ucla.edu

## Abstract

We study a random unitary circuit model of an impurity moving through a chaotic medium. The exchange of information between the medium and impurity is controlled by varying the velocity of the impurity, $v_d$, relative to the speed of information propagation within the medium, $v_B$. Above supersonic velocities, $v_d > v_B$, information cannot flow back to the impurity after it has moved into the medium, and the resulting dynamics are Markovian. Below supersonic velocities, $v_d < v_B$, the dynamics of the impurity and medium are non-Markovian, and information is able to flow back onto the impurity. We show the two regimes are separated by a continuous phase transition with exponents directly related to the diffusive spreading of operators in the medium. This is demonstrated by monitoring an out-of-time-order correlator (OTOC) in a scenario where the impurity is substituted at an intermediate time. During the Markovian phase, information from the medium cannot transfer onto the replaced impurity, manifesting in no significant operator development. Conversely, in the non-Markovian phase, we observe that operators acquire support on the newly introduced impurity. We also characterize the dynamics using the coherent information and provide two decoders which can efficiently probe the transition between Markovian and non-Markovian information flow. Our work demonstrates that Markovian and non-Markovian dynamics can be separated by a phase transition, and we propose an efficient protocol for observing this transition.

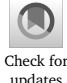

# 1 Introduction

The scrambling of quantum information is a concept crucial to a number of fields [1–9]. Its study helps quantify the complexity of quantum processes and can be related to the local recoverability of information [1, 10]. Quantum experiments carried out in reality are inextricable from their environments which often hinder information tasks when quantum information in the system is irretrievably lost to the environment [11–22]. However, there is always a finite probability that this information can flow back into the system and for certain physical processes this backflow is crucial. For example, in closed system thermalization the bulk of the system acts as an environment to, and becomes entangled with, local subsystems [23–25]. Furthermore, this information back flow, or non-Markovanity, may be a useful resource for quantum information processing [26, 27].

Recently, phase transitions in information dynamics have been demonstrated in a variety of settings ranging from PT-symmetric non-Hermitian systems [28–34] to systems monitored by projective measurements [35–44], to more general open systems involving qubits arranged in a variety of space-time geometries [14, 21, 32, 45–61]. It is therefore natural to wonder if open system dynamics undergo a phase transition in information flow as they are tuned between Markovian and non-Markovian limits. In this light, the authors of Refs. [62, 63] investigated non-Markovian effects of monitored systems with a non-Markovian environment but did not find a transition in non-Markovanity. In another context, Refs. [64, 65] found the non-Markovianity of an impurity is sensitive to a topological phase transition occurring within the environment. However, it is not clear if the two phases are distinguished by Markovian and non-Markovian dynamics. Furthermore, the environment in both systems are non-scrambling and Gaussian which is not characteristic of generic quantum systems.

Thus, in this work, we investigate the possibility of a phase transition in non-Markovianity of an impurity coupled to a quantum chaotic environment. We present a model in which the system can be taken between regions of zero and non-zero information backflow by only

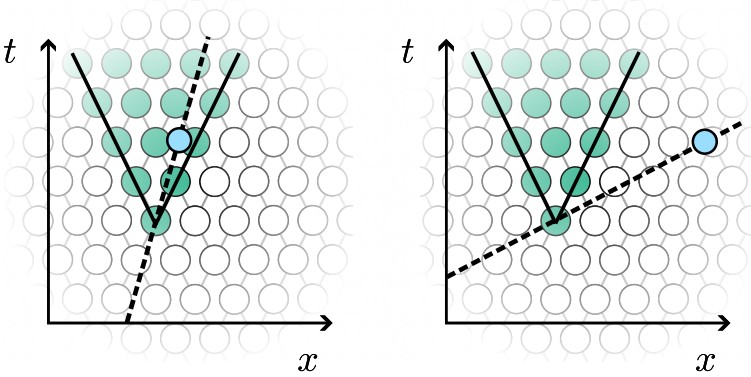

Figure 1: Cartoon of the model we consider: an impurity (blue circle) moving through a medium. The circles within the medium correspond to the unitary brick-work which drives the evolution of the medium. Initially, the medium contains no information about the impurity (clear circles). Then at a random time, the impurity interacts with the medium and leaks information into it at that particular point in spacetime. The information is rapidly scrambled away from this point (green circles) at a characteristic velocity. In this work, we investigate the behaviour for subsonic (left) and supersonic (right) impurity velocities.

varying the form of the coupling between the system and environment. Distinct from previous results [62, 64, 65], we find that these regions are separated by a continuous phase transition, and identify a critical point characterized by scale-free temporal correlations.

The model we consider describes an impurity moving through a chaotic medium. The capacity at which information can leave the impurity and re-enter at a later time can be tuned by varying the velocity of the impurity relative to the information scrambling velocity within the medium. We simulate the chaotic dynamics of the model using random unitary circuits and capture the flow of information by computing the out-of-time-order correlator (OTOC) [66–68]. The strength of the OTOC on the impurity at late times results from a combination of information retained by the impurity over time and information backflow from the medium. To isolate the contribution from backflow we enact a protocol in which the impurity is removed at an intermediate time and replaced with a fresh impurity. Any non-trivial operator support on the fresh impurity must be a result of feedback from the medium. Beginning with a single qubit impurity moving through a 1D medium, we find that on varying the velocity of the impurity a phase transition occurs between zero and non-zero backflow. Using a mapping between operator growth and a random walk [67, 68], we show that the criticality is a result of the diffusive growth of the operator within the medium.

The effect of varying the velocity of a local object, e.g. an impurity or local quench, within a medium relative to the speed of light within that medium has been studied previously in contexts such as Hawking radiation within many-body systems [69]. More generally the effect of this moving perturbation on the underlying state of the medium has been studied in a variety of approaches [70–74]. In contrast to these studies we instead focus on the dynamics of the information localized on the impurity and how this is affected by its velocity.

We then extend our model to that describing a 1D impurity moving through a 2D medium to study the effect of the backflow transition on scrambling within the impurity. In particular we focus on how the presence of backflow affects a distinct phase transition known to occur in the Markovian limit of this extended model [14]. This transition is between phases of persistent and vanishing scrambling within the impurity and we shall refer to it here as the

*scrambling transition*. We find that the scrambling transition is only possible when there is no operator backflow.

As in the scrambling transition, the backflow phase transition can also be observed in a particular channel capacity. The relevant channel in this case captures the information about the initial state contained within the medium and discarded impurity at late times. We show that a party with access to only these degrees of freedom can recover the initial impurity state with perfect fidelity when the channel capacity is maximal. This is achieved via a protocol where a fresh impurity is coupled to the medium and the dynamics are reversed. Additionally, we show that the backflow transition can be observed in a complementary channel. This channel characterizes the ability to decode information on the initial impurity state given access to its final state along with the initial states of the medium and the fresh impurity inserted in the reset step. The decoder for the complementary channel is reminiscent of that proposed in [10], for the Hayden-Preskill protocol. We associate these two quantum channels with distinct decoding protocols and in each case derive relations between the decoder fidelity and the channel capacity. In one case the decoder fidelity is optimal in the limit of zero backflow whereas for the other decoding it is enhanced in the opposite limit where non-Markovianity is maximal.

We conclude with a discussion of the distinction between the operator non-Markovianity introduced in this paper and non-Markovianity observed in time ordered correlations [75]. We highlight how operator non-Markovianity is only captured by protocols involving an echo such as the OTOC and decoder fidelities discussed in this paper. In contrast, the notions of Markovianity as discussed in Ref. [75], do not capture an echo and do not naturally capture the phase transition here. To connect the two we discuss a situation in which the observer has access to the initial environment but loses access at later times.

The paper is organized as follows. In Section 2 we introduce the model and give the details of the circuit implementation for both the 1D and 2D cases. We continue, in Section 3, to detail how we capture the operator dynamics and in particular the protocol we implement to capture operator backflow. First, we consider a single qubit impurity coupled to a 1D medium, and discuss the OTOC dynamics of this model in Section 4.1. Here we compute the degree of information backflow as a function of velocity and determine finite-time scaling exponents. In Section 4.2 we take a 1D impurity moving through a 2D medium and investigate the impact of the backflow transition on scrambling within the impurity. We then, in Section 5, investigate how the backflow transition is manifest in a pair of complementary channel capacities. We detail decoding protocols whose fidelity is directly related to these channel capacities and can be used to observe the transition in small, near term quantum computers. We conclude our discussion, in Section 6, where we emphasize what defines operator non-Markovianity before we finally, in Section 7, summarize our results and discuss potential future directions.

## 2   Model

Below we construct a random circuit model aimed at capturing the information dynamics of an impurity moving through a chaotic medium. To do so, we will consider the full unitary evolution of the total system of impurity and medium. By partitioning a closed system into subsystems labelled system and environment, we are then able to apply open systems concepts such as Markovianity. In this paper we always consider Markovianity of the impurity as a system with the medium as its environment. The main result is that the change from non-Markovian to Markovian operator dynamics on the impurity corresponds to a continuous phase transition.

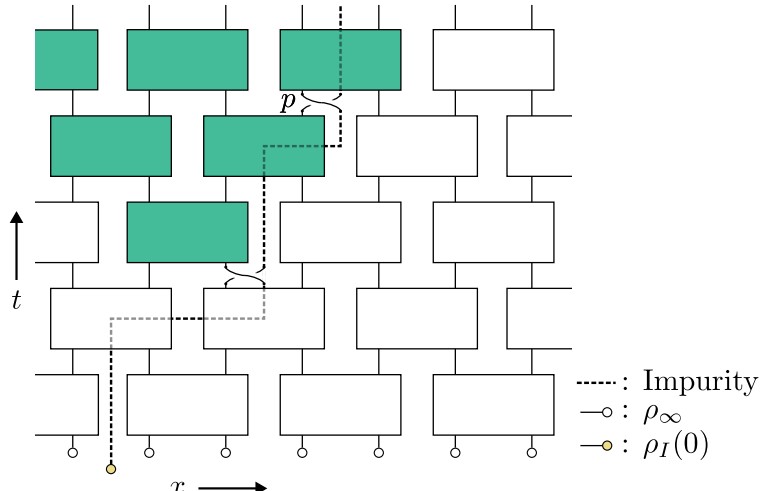

Figure 2: Circuit diagram of the model implementation. Here the medium qubits are all initialized in an infinite temperature state, $\rho_\infty$, and the impurity is initially in state $\rho_I(0)$. In this realization the impurity initially drifts two sites away from its origin and then interacts with the medium, after a further drift of one site it once again swaps qubits with the medium. The green gates indicate the lightcone associated with the information transferred into the medium by the first swap. The second swap is still within this scrambling lightcone and thus could lead to backflow of information from medium to impurity.

## 2.1 One-dimensional medium

Our first model is of a single impurity qubit moving over a 1D chain of qubits describing the medium. The chaotic evolution of the medium is captured by a random circuit and the impurity-medium interaction consists of swap gates applied at a fixed rate, $p$. We choose to use this form of interaction as it is analogous to the absorption/emission of quanta from/into the medium. However, we expect the results of this paper to hold for any interaction that results in operators being spread from impurity to medium and *vice versa*.[1] The circuit implementation of this evolution is depicted in Figure 2. The medium is initialized in an infinite temperature state and evolves under a brickwork of random unitaries drawn from the Clifford group. The Clifford group is a unitary 3-design: it exactly reproduces the first, second and third order moments of the Haar distribution [76]. Below, we consider the dynamics of OTOCs, which are second order functions of the brickwork unitaries, and hence are equivalent to those that would be generated by Haar random circuits.

Between every two layers of unitaries, the impurity qubit is swapped with the medium qubit at position $x \in \mathbb{Z}$ with probability $p$. Initially, the impurity is at position $x = 0$, and after each interaction step it shifts from $x$ to $x + d$ where the drift, $d \in \mathbb{Z}$, is drawn, independently at each step, from the following binomial distribution:

$$p(d) = \binom{d_{max}}{d} p_D^d (1 - p_D)^{d_{max} - d} . \tag{1}$$

Such a process implements a biased random walk with drift velocity $v_d = p_D d_{max} / \tau = p_D d_{max}$ where $\tau$ is the time between shifts. For the results presented in this article, we work in units of $\tau \equiv 1$ and fix $d_{max} = 20$.[2]

---

[1]We have confirmed this prediction numerically for an interaction consisting of random unitaries drawn from the Clifford group.

[2]We have confirmed that the main results of these paper do not qualitatively change if a deterministic shift is used instead.

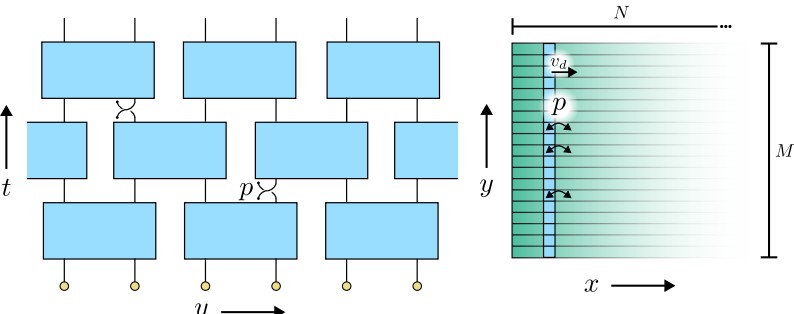

Figure 3: Illustration of the model implementation for a 1D impurity moving through a 2D medium. The left diagram shows the circuit which evolves the impurity through time with swaps occurring between each site and its independent strip of the medium. The right-hand cartoon shows how the impurity moves through the medium. Each green strip of the medium evolves independently via the circuit displayed in Figure 2.

### 2.1.1 Motivation and expectations

This model could describe, for instance, a mobile impurity immersed in a Bose gas, or equivalently a static impurity in a flowing gas [77–81]. In such a system, the general expectation is that information transferred from impurity to medium is rapidly scrambled away from the location of the interaction in a characteristic lightcone [67, 68]. This lightcone captures the velocity at which information is scrambled through the medium.

By tuning the impurity velocity relative to the speed of information scrambling within the medium, one may expect the information dynamics to undergo a transition. When the impurity moves slower than the information scrambling in the medium there is potential for information to flow back into the impurity and hence the reduced dynamics of the impurity can be non-Markovian. On the other hand, if the velocity is increased such that the trajectory falls outside the light cone, information backflow becomes impossible and the medium acts as a Markovian environment for the impurity. This limit, where the probability of feedback is zero, corresponds to $v_d > c$ where $c$ is the maximum velocity of information in the medium; in our case two layers of unitaries are applied per time step such that $c = 2$. In this limit the medium is effectively reset with respect to the dynamics of the impurity and there is no possibility of feedback. The general model and these two cases are depicted in Figure 1. In the Appendix A, we consider an alternate circuit where the scrambling within the medium only occurs up to the position of the impurity.

## 2.2 Two-dimensional medium

Motivated by Ref. [14] we also consider a chain of $N$ impurity qubits moving through a 2D lattice of $N \times M$ qubits. The 2D medium evolves as $N$ decoupled chains of $M$ qubits evolving under the brickwork circuit in Figure 2. The $N$ media are treated as independent but evolve under equivalent parameters i.e. $p$ and $v_d$ are global properties. Although treating the environments as independent is artificial it presents a simple limit in which we can explore the effect of backflow on the scrambling within the impurity. The 1D impurity chain takes steps through this medium in the $x$-direction with the step length being drawn from the distribution in Eq. (1). We allow for interactions to occur between the individual impurity sites via a random unitary brickwork equivalent to that occurring in the medium. Between applying each layer of unitaries to the impurity chain, each qubit of the chain is swapped independently into the medium with probability $p$. The 2D model is depicted in Figure 3; another perspective is gained by considering Figure 2 as a cross-section.

### 2.2.1 Motivation and expectations

For the 2D model we must account for the scrambling in the medium, the scrambling on the impurity, and the interaction between them. When taking the limit of Markovian impurity dynamics, $v_d > c$, the scrambling dynamics of the medium can be ignored and the circuit is equivalent to the one studied in Ref. [14]. There, the medium was a stationary reservoir of infinite temperature qubits. Similarly, in the limit $v_d > c$ the 1D impurity is deterministically beyond the scrambling lightcone and only interacts with infinite temperature qubits unaffected by past medium-impurity interactions. In this Markovian limit, the swaps occurring between the impurity and the medium suppress scrambling within the 1D impurity. In Ref. [14], it was shown that this suppression can result in a phase transition at a critical swap rate, $p_c$, above which scrambling in the impurity quickly vanishes corresponding the a complete loss of information into the environment. This scrambling transition was identified to be part of the directed percolation universality class [14].

Outside of the Markovian limit, $v_d < c$, information backflow is possible, and we investigate its effect on the scrambling transition identified in the Markovian limit. A simple expectation is that information backflow into the impurity increases information scrambling in the impurity and increases the critical swap rate.

## 3 Capturing operator dynamics and backflow

In this section we start with the 1D problem. We initialize an operator localized on the impurity, after some time on the order of $p^{-1}$ it is swapped into to the medium and then scrambled. To capture how the operator is scrambled within the medium we compute the out-of-time-order correlator (OTOC) defined by

$$C^{(M)}(x,t) = \frac{1}{4}\text{tr}\left\{\rho_0^{IM}\left[X^{(I)}(t), Y_x^{(M)}\right]^\dagger \left[X^{(I)}(t), Y_x^{(M)}\right]\right\}, \qquad (2)$$

where $X^{(I)}$ is the Pauli-X operator acting on the impurity and $X^{(I)}(t) = U_t^\dagger X^{(I)} U_t$ is this operator evolved to time $t$ in the Heisenberg picture. $Y_x^{(M)}$ is the Pauli-Y operator acting on the medium qubit at site $x$. OTOCs quantify where in Hilbert space a time-evolved operator has support. In addition to the operator weight in the medium, we capture the operator weight remaining on the impurity with the OTOC given by

$$C^{(I)}(t) = \frac{1}{4}\text{tr}\left\{\rho_0^{IM}\left[X^{(I)}(t), Y^{(I)}\right]^\dagger \left[X^{(I)}(t), Y^{(I)}\right]\right\}, \qquad (3)$$

where $Y^{(I)}$ is the Pauli-Y operator acting on the impurity. The choice of Pauli operators in the above expressions is arbitrary in our case; on averaging over Clifford circuit realizations, any two distinct non-identity Pauli operators for the initial and time-evolved operators would yield identical results.

If $C^{(I)}(t)$ vanishes at some time $t$ then the entire information content of the initial operator has flowed into the environment. In the case of Markovian environments this information is irretrievable unless one can access the environment. However, for a generic environment, information can flow back into the system at later times leading to non-Markovian revivals of the OTOC on the system. But this non-Markovianity is distinct from that typically considered in the field of open quantum systems which concerns the evolution and/or correlations of a state rather than an operator [75,82]. We will now outline a protocol in which we can measure the degree of this backflow.

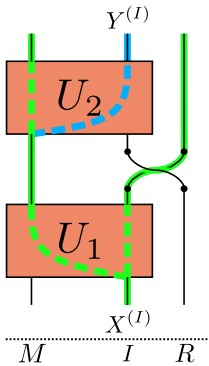

Figure 4: Schematic of the protocol used to measure the operator backflow quantified by $B = \frac{1}{4}\text{tr}\left\{\rho_0^{IM}[\tilde{X}^{(I)}(t), Y^{(I)}]^\dagger[\tilde{X}^{(I)}(t), Y^{(I)}]\right\}$. The total space is partitioned into the impurity, $I$, the medium, $M$, and the fresh impurity swapped in at the reset step, $R$. Highlighted in green and blue are the potential channels for information about $X^{(I)}$ to be carried, non-trivial operator weight on the impurity is detected by computing the overlap with $Y^{(I)}$. It is only possible for non-trivial weight to be carried to the fresh impurity via $M$ and channel highlighted in blue, with the perspective of the impurity as system and medium as environment this corresponds to a non-Markovian information flow. The unitaries, $U_1$ and $U_2$, are given by the circuit in Figure 2.

## 3.1 Operator backflow

We capture operator backflow with a protocol analogous to the causal break of the process tensor formalism [75, 83–88]. At the break the system is effectively reset such that across this break information can only be communicated via the environment; any dependence of the system's evolution post-break on its evolution pre-break is an indication of non-Markovianity. Our protocol in this spirit is depicted in Figure 4. First, the model is evolved for a time $t_1$, then the impurity is reset by swapping the qubit with one in an infinite temperature state. This fresh impurity is then evolved for a further time $t_2$ to give a total evolution time of $T \equiv t_1 + t_2$. After these steps we again track the operator weight on the impurity via an OTOC given by

$$B(v_d, T) = \frac{1}{4}\text{tr}\left\{\rho_0^{IM}\left[\tilde{X}^{(I)}(T), Y^{(I)}\right]^\dagger\left[\tilde{X}^{(I)}(T), Y^{(I)}\right]\right\}, \tag{4}$$

whose definition is equivalent to that of $C^{(I)}(T)$, but with the operators evolved under the backflow protocol, i.e. $\tilde{X}^{(I)}(T) = U_2^\dagger U_1^\dagger X^{(I)} U_1 U_2$. We enforce that $t_1, t_2 \gg p^{-1}$ such that the impurity and medium have significant time to interact before the reset step and subsequent OTOC readout.

In the case of random unitary evolution the environment strongly scrambles any information that flows into it and the system evolution is Markovian as characterized by a process tensor which factorizes between timesteps. However, it has recently been shown that although the state evolution may be Markovian or near-Markovian, the OTOC can still display non-Markovian correlations [89]. These correlations arise due to the Heisenberg evolution of the operator; in our model this causes the support of the operator to first spread onto the medium and then, after the reset step, onto the fresh impurity. This is highlighted in Figure 2 where we illustrate how information of the initial state can flow through the various channels of the process. The only way for this information to flow onto the final impurity is via the channel represented by the blue line. This corresponds to backflow of information from the medium via $U_2$, this information having flowed into the medium via $U_1$. We expect $U_1$ to transfer information from the impurity to the medium regardless of the impurity's velocity, but $U_2$ can only implement the reverse if the impurity remains causally connected to the initial transfer, i.e. if

it travels at sub-luminal velocities. Although we treat the full system unitarily, by considering the impurity sector as the system and the medium as the environment, we can consistently refer to the Markovianity of the reduced operator dynamics on the impurity.

## 4 Operator dynamics

In this section, we present the OTOC dynamics for both the 1D and 2D models introduced in Section 2. First, we investigate the dynamics of the backflow protocol for the 1D system and perform finite-time scaling on the late-time dynamics. For the 2D system, we consider the steady state of the OTOC without applying the reset step and investigate the effect of operator backflow on the scrambling of operators within the 1D impurity.

### 4.1 One-dimensional medium

We shall focus exclusively on $\overline{B(v_d, T)}$, the disorder-averaged weight of the OTOC on the impurity at time $T$ during the reset protocol depicted in Figure 4. In Figure 5 we plot $\overline{B(v_d, T)}$ at a late time as a function of the drift velocity $v_d$. A transition occurs between at a critical velocity of $v^* \approx 1.2$. To understand the significance of this value we must recall that the information dynamics within the medium is stochastic. So, while the rate at which the lightcone broadens is bounded from above by $c$, we can also associate a butterfly velocity, $v_B$, with the typical expansion rate. The average increase in the number of sites on which an operator has support after a time $t$ is then given by $2v_B t$ [68]. In our case we have $v_B = 1.2$, precisely the velocity at which we observe the transition.

In Figure 6 we show that the disorder-averaged backflow satisfies a finite-time scaling of the following form:

$$\overline{B(v_d, T)} = f\left((v_d - v^*)\sqrt{T}\right), \tag{5}$$

where $f$ is a universal scaling function. The origin of this exponent can be understood by analyzing the operator dynamics within the medium. Given that $t_2$ is much larger than the inverse swap rate $1/p$, we expect that, at $t_2$, the impurity has equilibrated with the local medium and will follow the dynamics of the nearest medium qubit. We are therefore interested in the weight of the OTOC in the medium at the position of the impurity; after disorder averaging that position is $\overline{x_I} = v_d t$.

To determine the OTOC at this position, we use the approach detailed in Ref. [68]. There, the average OTOC is considered by choosing the unitary bricks from the Clifford group and using the fact that the Clifford group is a 3-design [76]. This ensures the average over the Clifford group is the same as the average over the Haar group for the OTOC. Clifford circuits map Pauli strings to Pauli strings, and since we consider an initial Pauli $X$ operator, this operator remains a Pauli string for all time. From the definition of the OTOC we have $B(v_d, t) = 1$ if the operator $X^{(I)}(t)$ has character of $X$ or $Z$ on the impurity and otherwise equals zero. On disorder averaging if the operator has non-trivial (i.e. non-identity) support on a particular site then it has equal probability of having $X$, $Y$ or $Z$ character. Therefore we have that $\overline{B(v_d, t)} = \frac{2}{3}\overline{n_I(v_d, t)}$, where $n_I(v_d, t)$ is a density that is equal to one if $X^{(I)}(t)$ has non-trivial support on the impurity, and zero otherwise. We can equivalently express the disorder-averaged OTOC within the medium at position $x$ as $\overline{B^{(M)}(x, t)} = \frac{2}{3}\overline{n_M(x, t)}$, where $B^{(M)}$ is the OTOC within the medium during the backflow protocol. The late-time dynamics of $\overline{n_M(x, t)}$ is characterized by regions of zero and non-zero density separated by diffusively propagating boundaries. The dynamics of these boundaries can be well captured by a hydrodynamic description where the probability distribution of the boundary position evolves according to a Fokker-Planck equation [68]. The

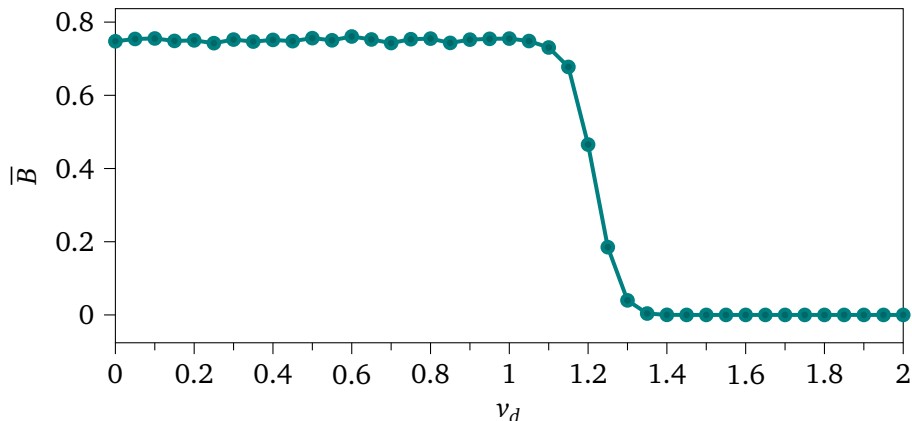

Figure 5: Backflow order parameter on variation of impurity drift velocity. Here we set $t_1 = 100$, $t_2 = 1000$ and $p = 0.1$. This is the result of an average over $10^4$ disorder realizations.

OTOC weight at the position of the impurity is given by

$$P(v_d, t) = \overline{n_M(\overline{x_I}, t)} = \frac{1}{2}\left[1 - \mathrm{erf}\left(\frac{v\sqrt{t}}{2\sqrt{D}}\right)\right], \tag{6}$$

where $v = v_d - v_B$ is the difference between the impurity velocity, $v_d$ and operator butterfly velocity $v_B$, and $D$ is a constant. This satisfies the diffusion-like scaling relation:

$$P(v, t) = F(v\sqrt{t}). \tag{7}$$

The backflow transition satisfies precisely the same scaling relation as that followed by the OTOC weight within the medium at the position of the impurity. This occurs because at late times the impurity matches the free dynamics of the OTOC within the medium without affecting it, the diffusive nature of this dynamics gives us the numerically observed scaling relations shown in Fig. 6. Moreover, the dynamics described by Eq. (6) are scale-free when $v = 0$. Away from this point the dynamics instead follow an error function which at long times corresponds to exponential relaxation on a timescale given by $\frac{v^2}{4D}$.

## 4.2 Two-dimensional medium

For the 2D problem the quantity of interest is the density of the OTOC over the entire impurity. We first extend the definition of the OTOC in Eq. 3 to have spatial dependence:

$$C^{(I)}(y, t) = \frac{1}{4}\mathrm{tr}\left\{\rho_0^{IM}\left[X_0^{(I)}(t), Y_y^{(I)}\right]^\dagger \left[X_0^{(I)}(t), Y_y^{(I)}\right]\right\}, \tag{8}$$

where $y$ has been introduced to index the position on the impurity. From this we can analogously define the, now site-dependent, density: $\overline{C^{(I)}(y, t)} = \frac{2}{3}\overline{n_I(y, t)}$. The site-dependence of the operator weight on the impurity is irrelevant to the question of backflow we study here. Instead, we consider the total operator weight

$$\overline{N_I(t)} = \frac{1}{N}\sum_y \overline{n_I(y, t)}, \tag{9}$$

where $N$ is the number of qubits within the impurity. This is the order parameter observing the phase transition in scrambling in the Markovian limit of this model [14]. The transition

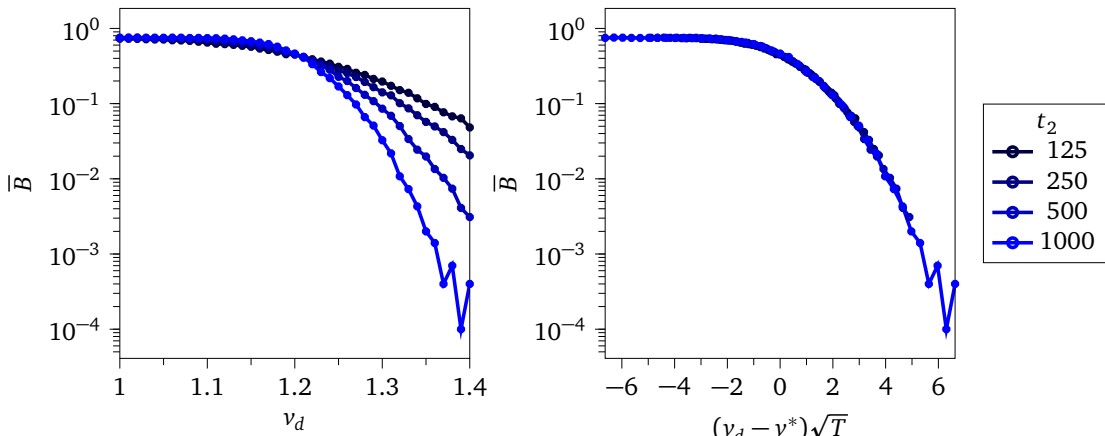

Figure 6: Left: Expanded view of the backflow phase transition near the critical point. Right: Data collapse under a finite-time rescaling. Here we set $t_1 = 100$ and $p = 0.1$. This is the result of an average over $10^4$ disorder realizations.

is characterized by the steady-state value of $\overline{N_I}$; below the critical swap rate $p_c$, $\overline{N_I}$ is finite and the system is in the scrambling phase, while above the crticial swap rate $\overline{N_I}$ vanishes and the system is in the "absorbing phase". This was shown to belong to the directed percolation universality class, where bond formation is promoted by the unitary gates and suppressed by the swap operations [14]. In the language of directed percolation, operator backflow can be associated with the formation of long-range temporal bonds in the temporal direction. We now consider the effect of these long-range temporal bonds on the percolation within the impurity.

Figure 7 shows the phase diagram we observe on varying the $p$ as we take $v_d$ across the backflow transition. There are four relevant regimes: subsonic velocities $v_d < v^*$, supersonic yet subluminal velocities $v^* < v_d \leq c$, superluminal velocities $v_d > c$, and a critical velocity $v_d = v^*$. In the subsonic regime, $v_d < v^*$, the operator flow is in non-Markovian phase of the 1D model. Here, operators flow back into the system and operator support on the impurity is always maintained; this region is purely scrambling. Conversely, in the superluminal regime, $v_d > c$, the impurity is beyond the deterministic lightcone of the medium and backflow is impossible. However, within this regime there is a region where results differ from the expected Markovian limit. This discrepancy can be attributed to the use of a stochastic drift velocity in our model. This introduces fluctuations, causing the impurity to occasionally traverse within the lightcone, thus deviating from the anticipated Markovian behavior. Despite this deviation, our model faithfully reproduces the Markovian outcomes as $v_d$ is increased, exhibiting two distinct phases: a percolating phase where the steady state value of $\overline{N_I(t)}$ is greater than zero and an absorbing phase where it vanishes. These phases survive even when $v^* < v_d < c$, but the critical swap rate, $p_c$, is shifted. The long-range bonds formed in this region are suppressed exponentially in time and their only affect is to renormalize the probability of the short-range bond formation. This renormalization diverges as we approach the critical velocity, $v_d = v^*$, and the probability of long-range bond formation becomes constant. This occurs because, at this velocity, the impurity is perfectly following the centre of the diffuse lightcone boundary within the medium such that from its reference frame the local operator weight is constant in time. Unless the long-range bonds are broken at a comparable, unphysical, rate then the impurity will always percolate.

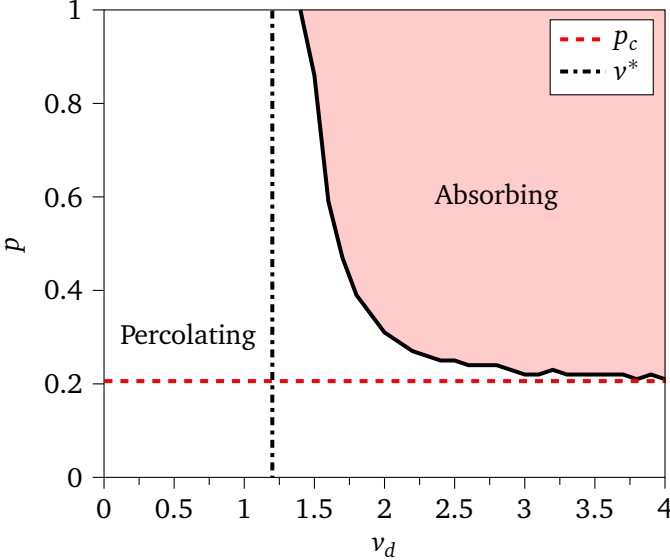

Figure 7: Phase diagram for the two-dimensional model as a function of swap rate, $p$, and impurity drift velocity, $v_d$. The absorbing phase corresponds to a steady state with $\overline{N_I} = 0$, while the percolating phase corresponds to $\overline{N_I} > 0$ in the steady state. The vertical line in the plot indicates the position of the critical velocity, $v^*$, of the backflow transition in the one-dimensional model. The horizontal line indicates the critical swap rate, $p_c$, of the scrambling phase transition in the Markovian model as reported in Ref. [14].

## 5 Information transition

We have shown that an impurity moving in a 1D medium can exhibit a phase transition in operator Markovianity by studying the OTOC dynamics of the system with the impurity reset operation shown in Fig. 4. Operator dynamics has been linked to the spread of information [1,14], and we will now explore how the phase transition in operator non-Markovianity manifests in the dynamics of quantum information. Specifically we study the coherent information transmitted through two complementary channels. These channels are those highlighted in the right-hand panel of Figure 8 as Yellow→ Blue for the echo protocol and Yellow→ Green for the Teleportation protocol. The results in this section were numerically simulated using the `QuantumClifford.jl` software package [90].

### 5.1 Echo protocol

In the thought experiment, Alice prepares the impurity qubit, $I$, in a particular state, $\rho_I$, and then allows it to move through the chaotic medium, $M$, evolving under the circuit described in Section 2. At an intermediate time she discards her qubit and replaces it with a maximally mixed qubit, $R$. At the end of the experiment, a second party, Bob, obtains access to both the discarded qubit, $D$, and the medium $M$. For clarity, we emphasise that Bob cannot access the impurity and has no knowledge of the input states.

We now show that the transition in operator backflow can also be observed in the information Bob has about the state, $\rho_I$, that Alice prepared in the impurity qubit at the beginning of the experiment. We quantify this information using the coherent information [91]. The coherent information is most easily calculated by introducing an ancilla qubit, $A$, that is maximally entangled with the initial state of Alice's qubit (the impurity). Conceptually, this ancilla

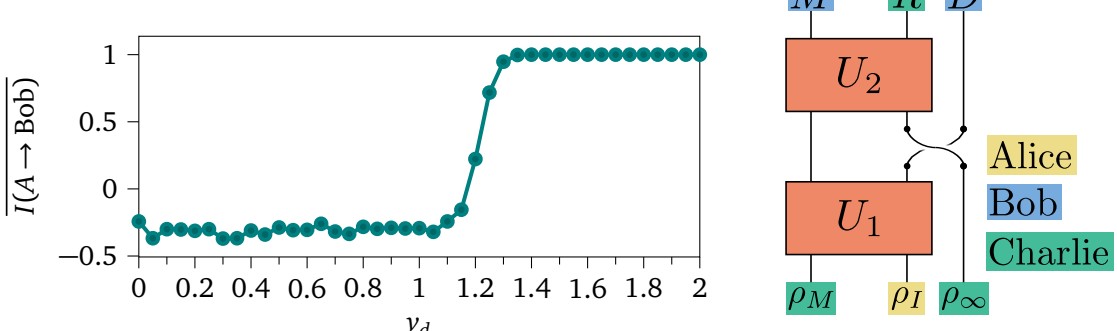

Figure 8: Left: Manifestation of backflow phase transition in coherent information. Here we set $t_1 = 100$, $t_2 = 1000$ and $p = 0.1$. This is the result of an average over 400 disorder realizations. Right: Diagram of the general protocol, the colors highlight the qubits accessible by the parties in both the echo and teleportation protocols. In the Markovian phase, there is no backflow and information of the initial state is only accessible via $M$ and $D$ so we expect Bob's decoder fidelity to be maximal. Conversely, Charlie's decoder is dependent on information flowing back onto the fresh impurity, $R$, and therefore we expect his decoder fidelity to be maximal in the non-Markovian phase.

acts a memory of the initial state of the impurity, and allows us to track how much of that information remains on the impurity as it interacts with the medium. Note that this protocol is complementary to the one in which Alice prepares a specific state $\rho_I$ on the impurity. The coherent information accessible by Bob is then given [91] as

$$I(A \rightarrow \text{Bob}) = S(\rho_{\text{Bob}}) - S(\rho_{\text{Bob} \cup A}), \tag{10}$$

where $S(\rho_C) = -\text{Tr}[\rho_C \log_2 \rho_C]$ is the von-Neumann entanglement entropy of the reduced density matrix of subspace $C$. The subspace labelled "Bob" contains the qubits that Bob can access, namely the qubit discarded by Alice and the final state of the medium: $\text{Bob} \equiv D \cup M$. For a circuit which allows Bob to recover Alice's state $\rho_I$ perfectly, his qubits will end up maximally entangled with the ancilla $A$ such that $I(A \rightarrow \text{Bob}) = 1$. While if Bob ends up with no information about Alice's state, his qubits and the ancilla will decouple, and $I(A \rightarrow \text{Bob}) \leq 0$.

During the first stage of the protocol, the initial state is entirely encoded in the impurity and the medium subspaces. Immediately after this impurity is discarded, prior to the occurrence of any swaps, this is still true and Bob has perfect access to Alice's initial qubit such that $I(A \rightarrow \text{Bob}) = 1$. As long as this remains true, Bob will always have complete information on the initial state. This is certainly true if the dynamics are Markovian. Here there is no operator back flow, the information remains in the medium and again Bob has perfect access to the information in Alice's initial qubit. Once the operator dynamics become non-Markovian, however, some information flows back into Alice's impurity and Bob loses the ability to decode the initial state of Alice's qubits. This is shown in the left-hand panel of Figure 8 where we plot $\overline{I(A \rightarrow \text{Bob})}$ as a function of $v_d$. There we see a transition occurring at the same critical velocity as the backflow transition observed by the OTOC. We again find the diffusive scaling exponents as shown in Figure 9, with the relevant scaling quantity being $1 - \overline{I(A \rightarrow \text{Bob})}$. This quantity is zero only in the region of zero backflow and increases as the velocity is decreased into the phase of non-zero backflow i.e. it acts as a witness of operator non-Markovianity.

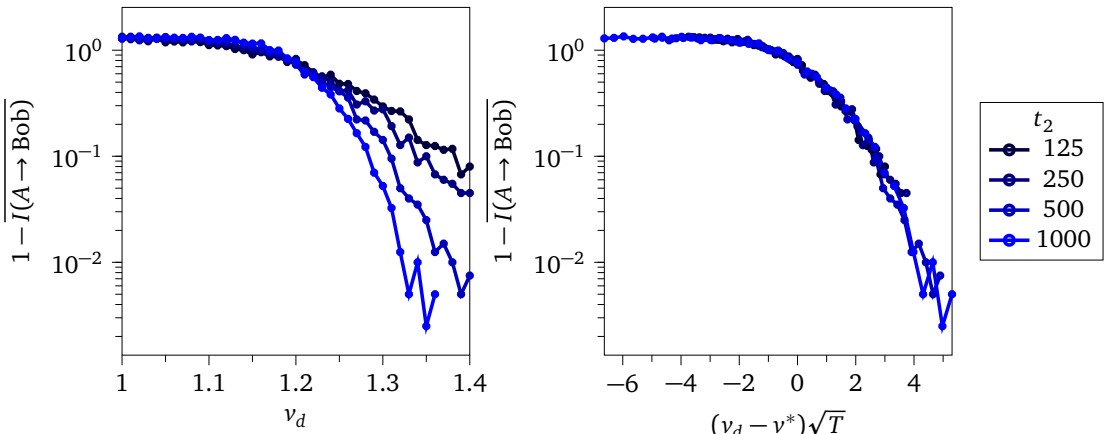

Figure 9: Left: Information phase transition around critical point. Right: Data collapse under a finite-time rescaling. Here we set $t_1 = 100$ and $p = 0.1$. This is the result of an average over 400 disorder realizations.

### 5.1.1 Echo decoder

The information accessible to Bob is maximal in the Markovian limit as all of the information about the Alice's qubit has flowed to Bob. We will now describe a decoder which yields a perfect fidelity in this limit and can be used to observe the transition. The decoder is depicted in Figure 10 as the green unitary operations. The construction is similar to the decoder in Ref. [14]; here Bob inserts a new impurity qubit, $I'$, prepared in an infinite temperature state and carries out the time-reversal of the evolution up to that point. The additional reset step in reverse consists of tracing out the impurity and replacing it with Alice's discarded impurity, $D$. The fidelity of this decoder is then given by the probability of a Bell pair forming between the final state of $D$ and $A$. Perfect fidelity is achieved when $\text{Tr}_R[U_2 \rho_R]$, and by extension $\text{Tr}_I[U_2^\dagger \rho_I]$, is unitary, which implies no information is transferred from $M$ to $R$. When this occurs, such as in the Markovian limit, $U_1^\dagger$ is trivially able to cancel the effect of $U_1$ and the initial state is recovered.

There is a connection between the coherent information and the decoder fidelity for a given circuit realization given by

$$I(A \rightarrow \text{Bob}) = 1 + \log_2 \mathcal{F}, \tag{11}$$

we derive this in Appendix B. In the supersonic phase the coherent information approaches unity and the decoder achieves perfect fidelity, while in the subsonic phase some information of the initial state flows back onto $R$ such that the decoder can no longer perfectly recover the initial state. We will now outline how a different protocol witnesses the same transition.

## 5.2 Teleportation protocol

Here we make use of a fundamental property of bipartite entropies to derive a distinct protocol which exhibits an equivalent transition in coherent information but for a channel complementary to the previous protocol. In this new protocol Alice carries out an equivalent role i.e. she prepares the impurity qubit, $I$, in a specific state, allows it to interact with the medium, $M$, and then replaces it with a new qubit, $R$, in an infinite temperature state. Throughout this the total evolution is unitary, as such any information which Bob cannot access from the discarded qubit, $D$, or the final state of $M$ must be accessible elsewhere in the system. With this in mind we now introduce a third party, Charlie, who is also attempting to deduce information about

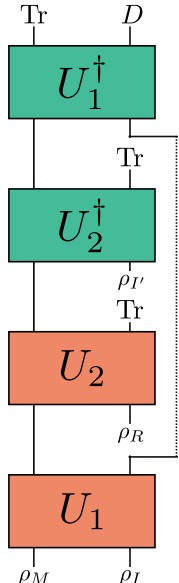

Figure 10: Diagram of the decoding protocol. When Alice reliquishes control of the impurity Bob performs a reversal of the dynamics up to that point except he only has access to the medium, $M$, and the impurity discarded by Alice in the refresh step, $D$.

the initial state of the impurity. Charlie has qubits that are maximally entangled with the initial states of $M$ and $R$, $A_M$ and $A_R$ respectively, as well as having access to the final state of $R$. These new ancilla, $A_M$ and $A_R$, play a similar role to $A$ in that they perfectly remember the initial state of the medium and the new qubit. Charlie having access to these implies he can make use of this information to help decode Alice's information. However, unlike Bob, Charlie has no knowledge of the discarded qubit, $D$, or the final state of $M$. The degrees of freedom Charlie can access are complementary to those accessible by Bob. This allows us to greatly simplify the calculation of the coherent information available to Charlie.

For pure states, like the one describing the quantum correlations between Alice, Charlie and Bob, all bipartite entanglement entropies satisfy

$$S(\rho_B) = S(\rho_{B\perp}), \tag{12}$$

where $B^\perp$ is the complement to the subspace $B$. Take a generic tripartite space consisting of $X \cup Y \cup Z$ and consider the coherent informations $I(X \to Y)$ and $I(X \to Z)$. Making use of Eq. (12) allows us to relate these in the following way:

$$\begin{aligned}
I(X \to Y) &= S(\rho_Y) - S(\rho_{X \cup Y}) \\
&= S(\rho_{Y\perp}) - S(\rho_{(X \cup Y)^\perp}) \\
&= S(\rho_{X \cup Z}) - S(\rho_Z) \\
&= -I(X \to Z).
\end{aligned} \tag{13}$$

Applying this relation to the relevant channels of Bob and Charlie then yields

$$I(A \to \text{Charlie}) = -I(A \to \text{Bob}). \tag{14}$$

This equivalence implies that $I(A \to \text{Charlie})$ must also witness the backflow transition when $\nu_d$ is varied. In contrast to Bob, Charlie has no information about Alice's qubits in the Markovian regime (when Bob can decode perfectly). While, in the regime of operator Non-Markovianity, Charlie has partial information about Alice's qubits.

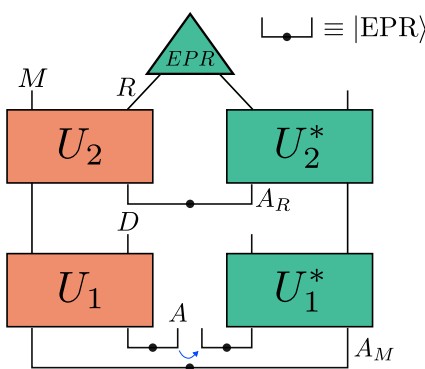

Figure 11: Diagrams depicting the complementary decoding protocol performed by Charlie given he only has access to the final state of the impurity along with the initial states of the medium and the impurity inserted at the reset state. The blue arrow indicates the goal of the protocol: to teleport information from the initial state of Alice's qubits to Charlie's ancillas.

### 5.2.1 Teleportation decoder

As before, we can construct a decoding protocol that reconstructs this partial information as shown in Figure 11. In that figure, the decoder is shown in green, and we show the diagram for the corresponding fidelity, $\tilde{\mathcal{F}}$, in Appendix C. There is a clear similarity with the Hayden-Preskill protocol decoder proposed by Yoshida and Kitaev [10]; in both cases a Bell measurement is performed to facilitate the teleportation of Alice's information to Charlie's qubits. The teleportation that Charlie hopes to achieve in this case is indicated by the blue arrow in Figure 11. The success of the teleportation improves when the bell measurement probability is lower (see Appendix C). Given that $R$ and $A_R$ are initially in a Bell state, the decoder has the worst fidelity when $\text{Tr}_M[U_2\rho_M(t_1)]$ is unitary, i.e. $U_2$ and $U_2^*$ do nothing to disentangle $R$ and $A_R$.

As in the previous protocol, we can relate Charlie's decoder fidelity, $\tilde{\mathcal{F}}$ to the coherent information (details in Appendix C):

$$I(A \to \text{Charlie}) = 1 + \log_2 \tilde{\mathcal{F}}. \tag{15}$$

The logarithm of the fidelity, therefore also observes the backflow transition. Note, that while Bob's decoder works perfectly in the Markovian phase, Charlie's decoder only partially succeeds in the non-Markovian phase. This is the main difference with the Hayden-Preskill protocol, and highlights the fact that the scrambling of operators back on to the impurity is not maximal, as is assumed for the black hole in the Hayden-Preskill protocol.

The phase transition in the OTOC dynamics explored in Section 4.1 is related to a revival in operator support on the impurity due to non-Markovian feedback from the medium. However, the non-Markovian effects only arise because of the echo step carried out in the OTOC calculation. This new protocol we have identified undergoes a phase transition in non-Markovianity more aligned with its traditional definition: in terms of the backflow of information from environment to system. While the decoding is facilitated by Charlie initially having access to the medium, this access is lost during the dynamics and the enhancement of the decoder fidelity is driven by feedback of information onto the impurity.

## 6 Operator Markovianty

In this paper we found a model which shows a phase transition in the operator dynamics of an impurity moving through a medium. At slow velocities, the operator dynamics were

non-Markovian, while at high velocities the operators dynamics were Markovian. Above, we introduced several protocols that observe the transition, and in this section, we compare non-Markovianity in the operator dynamics to non-Markovianity in time ordered correlations. We capture non-Markovianity in the operator dynamics explicitly with the backflow OTOC introduced in Eq. (4) and Figure 4. The back flow OTOC determines the weight of an operator, initially localized on the impurity, at a late time $T$ on the impurity, given that the impurity was replaced at an earlier time $t_1$. The replacement removes all operator support from the impurity such that any subsequent operator support there can only be due to operator support in the environment, which itself must have flowed from the initial impurity, spreading back into the fresh impurity.

While this back flow OTOC detects non-Markovianity in the dynamics of the operator, it does not detect the existence of non-Markovianity in time-ordered correlations. Non-Markovianity in time-ordered correlations can instead be characterized by the operational framework introduced in Ref [75]. Considering non-Markovianity of the impurity in that framework, we must assume that the medium is operationally in accessible. This implies that an echo of the environment cannot be performed and the backflow OTOC cannot be captured by that framework. This also allows for the possibility that the time-ordered correlations do not show non-Markovianity, but the backflow OTOC does. This is likely the case for the dynamics discussed here when the environment is initialized in a maximally mixed state. Even though operators flow back onto the system, they also have large support in the medium, and thus imply large correlations between the system and the environment. Since the chaotic dynamics of the medium spread these correlations non-locally, they will be inaccessible to the local probes provided by the multi-time correlations of the impurity.

Nonetheless, there is still a notion in which the information dynamics is non-Markovian. This is demonstrated by the two channel capacities and decoders discussed above. For the echo decoder, introduced in Section 5.1, the information accessible to Bob is used to decode Alice's qubit. In the Markovian phase, Bob obtains all information about Alice's qubit and can perfectly decode. In contrast, when operators spread back on to the impurity after $t_1$, some of the information is lost to him and he loses some ability to decode. Note that while some information about Alice's qubits has left the medium (Bob's qubits), it has not completely transferred into the impurity. Instead, that information has spread into the correlations between Bob's qubits and the fresh impurity. In both phases, the late-time impurity can not recover Alice's qubit, while only in the Markovian phase, when Bob doesn't lose information due to operator backflow onto the impurity, can Bob decode.

A different reasoning applies to the teleportation decoder discussed in Section 5.2. In contrast to the echo decoder, Charlie requires access to the initial state of the medium, but not the final state. If we consider the medium to be inaccessible at all times such a protocol falls outside the operational notion of non-Markovianity as discussed in Ref [75]. However, if we consider an experiment to have initial access, but to lose that access after the first step of dynamics we match the two pictures. In this case, the fidelity of the teleportation decoder has the form of a multi-time correlation. In the Markovian phase, Charlie cannot decode, and the fidelity vanishes, while in the non-Markovian phase, the teleportation protocol partially succeeds and the fidelity becomes positive.

# 7 Conclusion

Within the framework of random unitary circuits we have studied an impurity moving through a chaotic medium and the flow of information between them as characterized by an OTOC. Information deposited into the medium is rapidly scrambled away from the deposition site at

the speed of sound $v_B$. We show that the supersonic ($v > v_B$) and subsonic ($v < v_B$) impurity velocities correspond to regimes of zero and non-zero operator backflow respectively. This was determined by defining a protocol in which the impurity qubit was swapped with a fresh one uncorrelated with the environment and computing the OTOC on this new qubit. The scaling exponents associated with this transition was shown to be related to the diffusive evolution of the OTOC in the medium.

We considered the implications of the backflow transition on the scrambling transition previously observed in a 1D impurity interacting with a Markovian environment [14]. As expected from that work, this transition is only possible at supersonic velocities where the backflow occurs with a rate which decays exponentially with a characteristic timescale. When this is no longer the case, i.e. when the probability of backflow remains constant or decays algebraically with a small exponent as in Appendix A, there can be no absorbing phase. However, in the case of power-law decay, there is a potential for the transition to survive, provided the power is sufficiently large [92].

We went on to show how the operator non-Markovanity transition is also manifest in a quantum channel capacity, namely that between the initial impurity state and the final state of the environment plus the qubit discarded in the backflow protocol. We associated with this channel a decoder whose fidelity was closely related to the channel capacity and became ideal in the Markovian limit. By considering the complement to this quantum channel we then derived an alternative protocol which resembles the proposed in [10] for the Hayden-Preskill protocol. In this case the performance of the decoder was tied to the degree of backflow from the environment becoming optimal in the non-Markovian limit.

We have presented a relatively simple model that undergoes a phase transition in information backflow. Similar to the scrambling transition [14] and the measurement induced phase transition (MIPT) [35,36,93,94], the transition occurs in the dynamics of information. Unlike the MIPT, but similar to the scrambling transition, there is no exponential time complexity barrier in observing the backflow transition. Similar to the scrambling transition, the dynamics of information in the backflow transition can be observed using one of the two simple decoder shown above. The main gain in comparison to the scrambling transition (which requires storing $L^2$ qubits of the medium) is that the backflow transition occurs in 1D medium, and only requires storing $L$ qubits during the time evolution and decoding. This difference may be of importance when simulating the transitions on near term quantum computers.

Beyond realizing this transition on modern near-term devices other promising directions would be to extend this to a more realistic open quantum model such as a spin-boson type model [95, 96] or a Kondo model [97]. Already we can draw some conclusions when considering a mobile impurity coupled non-chaotic environments such as these. A typical example is a mobile atom coupled to electromagnetic radiation. In this case, the impurity is not only Markovian if it moves outside the light cone, but can also be if it moves inside the light cone. This is because operators dynamics are ballistic in a wave medium [98], and operators only have significant support close to the light cone. This is in contrast to a generic chaotic medium, where operators have non-negligible support through out the light cone. In studying open quantum systems closer to reality, recent techniques developed to treat non-Markovian dynamics will be of great assistance [99–103].

## Acknowledgments

The authors gratefully acknowledge the computing time granted on the supercomputer MOGON 2 at Johannes Gutenberg-University Mainz (hpc.uni-mainz.de).

**Funding information** This work has been funded by the Deutsche Forschungsgemeinschaft (DFG, German Research Foundation) - TRR 288 - 422213477 (project B09), TRR 306 QuCoL-iMa ("Quantum Cooperativity of Light and Matter"), Project-ID 429529648 (project D04) and in part by the QuantERA II Programme that has received funding from the European Union's Horizon 2020 research and innovation programme under Grant Agreement No 101017733 ("QuSiED") and by the DFG (project number 499037529), and by the Dynamics and Topology Centre funded by the State of Rhineland Palatinate.

# A Alternate 1D circuit

Here we show that the backflow transition, with modified exponents, also occurs when the 1D model of the main text is slightly modified. We now allow for scrambling to only occur up to the position of the impurity as depicted in Figure 12. In this case the backflow order parameter on variation of $v_d$ is shown in Figure 13.

The change to the circuit modifies the spread of the operator within the environment and in particular the weight of the OTOC at the impurity position which is instead given by

$$P_1(v_d, t) = \frac{\exp\left(\frac{-v^2 t}{4D}\right)}{\sqrt{\pi D t}\left[1 + \text{erf}\left(\frac{v\sqrt{t}}{2\sqrt{D}}\right)\right]}. \tag{A.1}$$

This in turn satisfies a distinct scaling relation:

$$P_1(v, t) = \sqrt{t}F_1(v\sqrt{t}), \tag{A.2}$$

and we see this manifest as well in the scaling of the backflow order parameter in Figure 14.

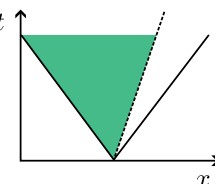

Figure 12: Cartoon of the modified boundary conidition where scrambling in the environment, as indicated by the green region, can only occur up to the position of the impurity, the dashed line.

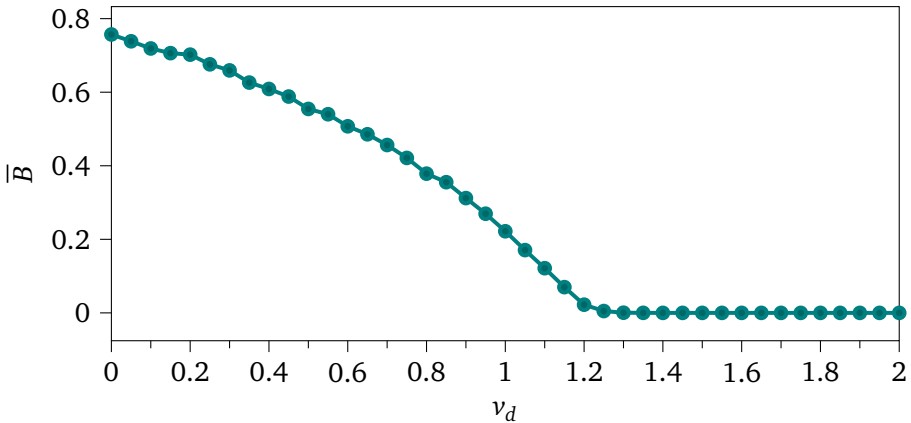

Figure 13: Backflow order parameter for the alternate circuit on variation of impurity drift velocity. Here we set $t_1 = 100$, $t_2 = 1000$ and $p = 0.1$. This is the result of an average over $10^4$ disorder realizations.

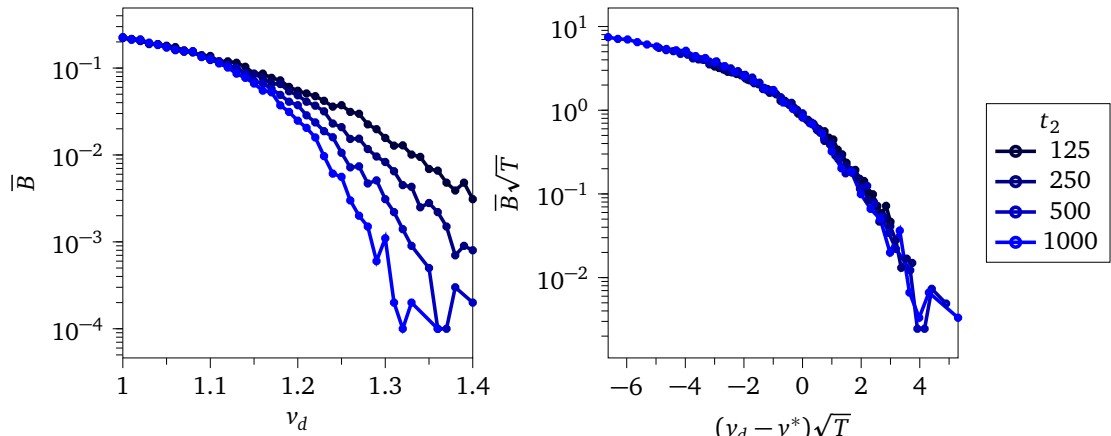

Figure 14: Left: Expanded view of the backflow phase transition for the alternate circuit near the critical point. Right: Data collapse under a finite-time rescaling. Here we set $t_1 = 100$ and $p = 0.1$. This is the result of an average over $10^4$ disorder realizations.

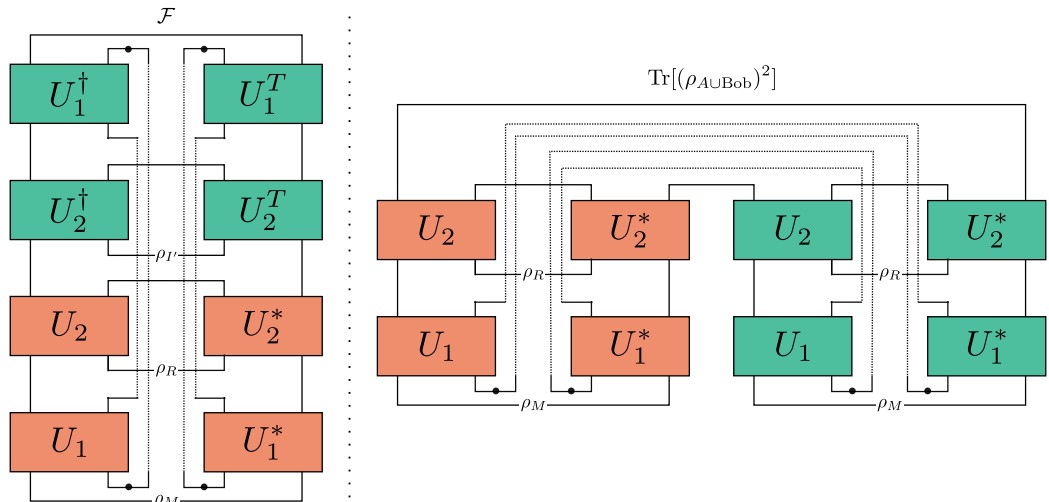

Figure 15: Left: The fidelity of Bob's decoding protocol for a given unitary circuit, $U$, and initial states: $\rho_M$, $\rho_R$ and $\rho_{I'}$. Right: The purity of the reduced density matrix $\rho_{A\cup\text{Bob}}$. If $\rho_M$, $\rho_{I'}$ and $\rho_R$ are all infinite temperature states then these two diagrams are equivalent down to a numerical factor.

## B  Derivation of Eq. (11)

The fidelity of the decoder for Bob's protocol is related to the purity of the reduced density matrix $\rho^{A\cup\text{Bob}}$ prior to decoding via

$$\mathcal{F} = d_M^{-1}\text{Tr}(\rho^2_{A\cup\text{Bob}}), \tag{B.1}$$

where $d_M$ is the Hilbert space dimension of the medium; we show this diagrammatically in Figure 15. By manipulating the fidelity diagram the decoder becomes a replica in the purity diagram. These two diagrams correspond when $\rho_M$, $\rho_{I'}$ and $\rho_R$ are infinite temperature states. Infinite temperature states in the fidelity are related to trace operations in the purity (and *vice versa*) via

$$\rho_B^\infty \leftrightarrow d_B^{-1}\text{Tr}_B[\cdot], \tag{B.2}$$

where $d_B$ is the Hilbert space dimension of $B$. These dimensional factors mostly cancel in our case except for $d_M$.

Now, the entanglement spectrum for Clifford circuits is flat[]; this results in an equivalence between Renyi entropies, $S_n(\rho_C)$, defined by

$$S_n(\rho_C) = \frac{1}{n-1}\log_2\text{Tr}[\rho_C^n]. \tag{B.3}$$

This combined with the relation $S(\rho_C) = \lim_{n\to 1}S_n(\rho_C)$ allows us to make the following substitutions:

$$\begin{aligned}
I(A\to\text{Bob}) &= S_2(\rho_\text{Bob}) - S_2(\rho_{A\cup\text{Bob}})\\
&= -\log_2\text{Tr}[\rho_\text{Bob}^2] + \log_2\text{Tr}[\rho_{A\cup\text{Bob}}^2]\\
&= 1 + \log_2\mathcal{F},
\end{aligned} \tag{B.4}$$

where to arrive at the last line we used $S_2(\rho_\text{Bob}) = -\log_2(d_M d_I) = 1 - \log_2(d_M)$.

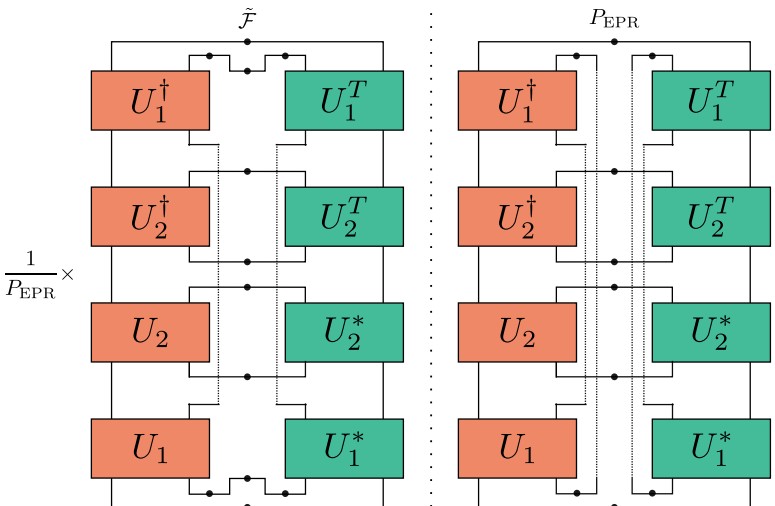

Figure 16: The fidelity of the second Hayden-Preskill-like protocol of the main text. The numerator trivially contracts to yield a constant factor and the non-trivial dependence on the unitary appears in the probability of measuring a bell pair, $P_{\text{EPR}}$.

## C  Details of HP fidelity and derivation of Eq. (15)

The fidelity for Charlie's decoding protocol is shown diagrammatically in Figure 16 where the only non-trivial factor is given by the probability of measuring a Bell pair, $P_{\text{EPR}}$. By comparing the diagram for $P_{\text{EPR}}$ and that for the original decoder's fidelity, $\mathcal{F}$ in Figure 15, it is apparent that

$$P_{\text{EPR}} \equiv \mathcal{F}. \tag{C.1}$$

The numerator in $\tilde{\mathcal{F}}$ trivally contracts to yield a factor of $d_I^{-2}$ where $d_I$ is the Hilbert space dimension of the impurity. Altogether we have the following relation between the two fidelities:

$$\mathcal{F} = \frac{1}{\tilde{\mathcal{F}} d_I^2}. \tag{C.2}$$

Substituting this into Eq. (11) and making use of Eq. (14) (along with $d_I = 2$) then yields Eq. (15).

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
