# Peer review of "Markovian to non-Markovian phase transition in the operator dynamics of a mobile impurity"

_SciPost Physics Core, doi:SciPost Phys. Core 7, 060 (2024)_

## Round 1 · Referee Report · Anonymous (Referee 1) · 2024-3-1

Report

I have studied the submitted manuscript “Markovian to non-Markovian phase transition in the operator dynamics of a mobile impurity”. The authors are interested in the dynamics of a mobile impurity dragged with velocity v, in a bath with maximum butterfly velocity vb: the main question concerns the Markovianity of the bath on the impurity. In other words, whether it is possible that excitations, or more generally information, triggered by the impurity can travel back into it at a later time.
As one can naively expect from the interplay of v and vb, if the impurity is superluminal v>vb, the bath cannot keep up with the impurity and the effective dynamics of the latter is Markovian.
The authors then use various tools of quantum information to quantify this phenomenon.

I think the physics is clearly explained in the manuscript and I do not have substantial questions to ask the authors, although I have a main concern: the scenario and somewhat the ideas investigated by the authors are not new, but already present in literature that it has not been properly acknowledged.

For example, there are at least the following papers that use the interplay of the velocity of an impurity and the background maximum velocity to investigate various aspects of transport and information propagation (and I leave aside the huge literature on impurities in cold atoms, see eg. Schecter, Gangardt, Kamenev, New J. Phys. 18 065002 (2016) and refs citing it)

Agarwal, Bhatt, Sondhi, PRL 120, 210604 (2018)
Bastianello, De Luca, PRL 120 (6), 060602 (2018)
Bastianello, De Luca, PRB 98 (6), 064304 (2018)
Mitra, Ippoliti, Bhatt, Sondhi, Agarwal, PRB 99, 104308 (2019)
De Luca, Bastianello, PRB 101 (8), 085139 (2020)
Maertens, Bultinck, Van Acoleyen, PRB 109, 014309 (2024)

This is surely a non-exhaustive list. In particular, the authors should be aware of PRB101 where very similar questions to those investigated in the submitted manuscript have been studied: Fig. 1 of the mentioned ref is essentially Fig. 1 of the submitted manuscript, then in section IV of PRB101 a manifestation of the “Markovianity” discussed by the authors is described, and notice that both Hamiltonian systems and circuits are discussed in PRB101. The authors’ work is original in the perspective and tools used to investigate this scenario, but due to the similarities with the mentioned work it would be nice if the authors could connect with it.

Overall, I think the problem studied by the authors is of some interest and the manuscript is accessible to non-experts, but in my opinion the manuscript does not contain the substantial novelty required by the high standards of Scipost Physics. Hence, I look at Scipost Core as a more proper journal to convey these findings and there I can endorse the publication of this manuscript, provided my concerns are taken into account.

Requested changes

See Report

  • validity: high
  • significance: good
  • originality: good
  • clarity: high
  • formatting: perfect
  • grammar: perfect

Author:  Dominic Gribben  on 2024-06-28  [id 4594]

(in reply to Report 1 on 2024-03-01)

Please find our response attached.

Attachment:

Referee_response_aZcCjWt.pdf

---

## Round 1 · Referee Report · Anonymous (Referee 2) · 2024-3-7

Report

This manuscript is about toy models of a mobile impurity interacting with a chaotic medium. The focus is on the flow of quantum information between them. The main message is that there are two phases: when the impurity velocity is greater/lesser than the “butterfly velocity” (the effective maximum speed of information spreading) in the medium, information cannot/can flow from the impurity to the system and back. Hence the effective dynamics of the impurity is Markovian/non Markovian. The authors aimed to characterise the transition between them, using an “operator back flow ” out-of-time order correlator (which quantifies how much information flows from the impurity to the medium and back). They also studied a variant model with a 1D impurity moving in a 2D environment; this part is also motivated by a previous work on a model of operator growth with swapping, which exhibits a directed percolation vs absorption transition. The current 2D model extends the phase diagram into a 2-parameter space. Finally the authors considered the information transmission in terms of quantum channels, and constructed decoders à la Hayden Preskill.

The manuscript is overall well written, clear and accessible.

To be honest, I am more impressed by the breadth of the work than the depth of its main finding, which, as summarised above, is rather unsurprising. The technical setup of Clifford brickwork + OTOC is rather formal from a physical viewpoint, and not ground-breaking in terms of mathematical craft. More importantly, I find the study design suffers from two shortcomings:

  • The impurity-medium interaction is modelled by a swap with probability $p$, and hence the fluctuation generated is stochastic (averaging over realisations). This choice looks quite artificial and arbitrary, and is not motivated or justified. Even within the Clifford framework there are many other possible choices. Their potential effect on the results was not sufficiently addressed.

  • If I guessed the meaning of eq (1) correctly (see below), the impurity performs a biased random walk. In my opinion this is an unfortunate choice. Indeed, the diffusive critical scaling could be also attributed to the stochastic diffusive broadening of the random walk, as well as the quantum butterfly front broadening. Therefore, the definition of the model introduces artificial noise that obfuscates the critical physics it tries to describe.

Finally, since the transition is interpreted as one between Markovian and non Markovian open dynamics, it would be more convincing to provide an effective description of the impurity dynamics, at least in the Markov phase.

In summary I find the manuscript publishable (after suitable revision, see below), but not significant enough for SciPost. SciPost Core may be a more suitable venue.

There are also a number of minor issues/questions:

  • In the “model” section, the definition of the models and the discussion on the physical motivation + anticipated results are too much intertwined. For the reader’s convenience I find it preferable to restructure the section, and separate the definition from the discussion.

  • Around eq (1) it should be clearly stated whether d is independently drawn for each time step (see above).

  • The paragraph below eq (1), where both $v_B$ and $c$ appeared, can be quite confusing (independently of the above point). If both veloicities have to be introduced here, it will be helpful to state that $c = 2$ is a strict upper bound for $v_B$, but $v_B = 1.2$ was known to be smaller than $c$. In fact I feel that it is better to postpone the introduction of $v_B$ to the results section, where it truly belongs.

  • In Section 3, it can be a bit confusing that the basic OTOCs (2) and (3) are defined but somehow discarded/amended (in section 3.1) by more a more complex protocol. The latter deserves to be better motivated than essentially quoting several references. The OTOC $B(v_B, t)$ could be more clearly defined (since that is the one that will be used).

  • Related, in figure 4, which illustrates $B(v_B, t)$, it will be helpful to mark where there the involved operators act. Instead, information about Alice, Bob and Charlie is useful only in Section 5. So a figure showing their roles should rather accompany Figure 8.

  • In the conclusion, the authors stated that “Unlike the MIPT, but similar to the scrambling transition, this transition does not require the post selection of measurement outcomes.” This phrase can be misleading. The standard MIPT is formulated without post selection, that is, upon averaging over outcomes. Observing MIPT requires obtaining the same outcome many times or overcoming the sampling problem in another way. But these two facts should be clearly distinguished.

Requested changes

see above

  • validity: high
  • significance: good
  • originality: good
  • clarity: good
  • formatting: excellent
  • grammar: perfect

Author:  Dominic Gribben  on 2024-06-28  [id 4593]

(in reply to Report 2 on 2024-03-07)

Please find our response with figures attached.

Attachment:

Referee_response.pdf

---

## Round 2 · Referee Report · Anonymous (Referee 1) · 2024-7-11

Report

In my previous report, I addressed two main concerns: i) the preexisting literature was not properly referenced, and ii) while the study and result are interesting, I think they do not meet the high standard of SciPost Physics and I recommended SciPost Physics Core as a more proper venue.
Both issues have been resolved by the authors, and therefore I can recommend the manuscript for publication in SciPost Physics Core.

Recommendation

Publish (meets expectations and criteria for this Journal)

---

## Round 2 · Referee Report · Anonymous (Referee 2) · 2024-7-18

Report

I thank the authors for considering the referees' concerns, which are adequately addressed by the revised manuscript. I support its publication in SciPost Physics Core.

Recommendation

Publish (easily meets expectations and criteria for this Journal; among top 50%)

---

## Editorial Decision

published